

# Alpha lipoic acid mitigates adverse impacts of drought stress on growth and yield of mungbean: photosynthetic pigments, and antioxidative defense mechanism

Naima Hafeez Mian, Muhammad Azeem, Qasim Ali, Saqib Mahmood and Muhammad Sohail Akram

Government College University, Faisalabad, Faisalabad, Pakistan

## ABSTRACT

**Context:** Exogenous use of potential organic compounds through different modes is a promising strategy for the induction of water stress tolerance in crop plants for better yield.

**Aims:** The present study aimed to explore the potential role of alpha-lipoic acid (ALA) in inducing water stress tolerance in mungbean lines when applied exogenously through various modes.

**Methods:** The experiment was conducted in a field with a split-plot arrangement, having three replicates for each treatment. Two irrigation regimes, including normal and reduced irrigation, were applied. The plants allocated to reduced irrigation were watered only at the reproductive stage. Three levels of ALA (0, 0.1, 0.15 mM) were applied through different modes (seed priming, foliar or priming+foliar).

**Key results:** ALA treatment through different modes manifested higher growth under reduced irrigation (water stress) and normal irrigation. Compared to the other two modes, the application of ALA as seed priming was found more effective in ameliorating the adverse impacts of water stress on growth and yield associated with their better content of leaf photosynthetic pigments, maintenance of plant water relations, levels of non-enzymatic antioxidants, improved activities of enzymatic antioxidants, and decreased lipid peroxidation and $H_2O_2$ levels. The maximum increase in shoot fresh weight (29% and 28%), shoot dry weight (27% and 24%), 100-grain weight (24% and 23%) and total grain yield (20% and 21%) in water-stressed mungbean plants of line 16003 and 16004, respectively, was recorded due to ALA seed priming than other modes of applications.

**Conclusions:** Conclusively, 0.1 and 0.15 mM levels of ALA as seed priming were found to reduce the adverse impact of water stress on mungbean yield that was associated with improved physio-biochemical mechanisms.

**Implications:** The findings of the study will be helpful for the agriculturalists working in arid and semi-arid regions to obtain a better yield of mungbean that will be helpful to fulfill the food demand in those areas to some extent.

Corresponding author
Muhammad Azeem,
mazeem@gcuf.edu.pk

## INTRODUCTION

*Vigna radiata* L. (Wilczek) (commonly named mungbean) is known as golden or green gram because of the high protein content in its seeds, which are used highly in sprouted or dry seed form. Mungbean is the third most crucial legume grain after pigeon pea and chickpea (*Bangar et al., 2019*). It is cultivated predominantly across Asian countries and expanded to some parts of South America, Australia, and Africa (*Nair et al., 2019*; *Pratap et al., 2020*). It is a fast-growing, self-pollinating, diploid, and short-duration crop. It is helpful as the effective utilization of summer fellows to increase crop production and cropping intensity (*Singh et al., 2016*). *Vigna radiata* L. (Wilczek) seeds contain antifungal, antibacterial, and anticancer properties (*Turk et al., 2018*; *Uppalwar, Garg & Dutt, 2020*). Mungbean has low input requirements and wider adaptability (*Khalid et al., 2019*; *Singh et al., 2022*). As a leguminous crop, it has a robust root system helpful in the atmospheric nitrogen fixation in soil (about 58–109 kg/ha) with *Rhizobium* (*Allito, Nana & Alemneh, 2015*; *Lindstrom & Mousavi, 2020*). Therefore, it is essential in sustaining productivity and improving soil fertility (*Favero et al., 2021*). It is an excellent source of antioxidants like phenolic and flavonoids. It is also a rich source of micronutrients and vegetable proteins (*Guo et al., 2012*; *Nair et al., 2015*; *Foyer et al., 2016*) and hence has been used in multifarious food (*Arnoldi et al., 2014*; *Ebert, 2014*) and green gram manure (*Boelt et al., 2014*). Water stress is one of the principal environmental stresses that hinders plant growth, development, and crop productivity, especially in arid and semi-arid environments (*dos Santos et al., 2022*).

The deleterious impacts of water stress include a reduction in plant growth, disruption of photosynthetic pigments, reduction in water and nitrogen use efficiency, and abnormalities in cell structure. It also modifies the activities of cellular metabolites (*Chen et al., 2019*). Moreover, water stress causes over-accumulation of reactive oxygen species (ROS), resulting in oxidative damage and many adverse impacts, including stomatal closure and altered activities of cellular enzymes, resulting in reduced photosynthesis. The generation of ROS also causes membrane lipid peroxidation and subsequently damages the membrane, proteins, and nucleic acids (*Hasanuzzaman et al., 2021*).

Plants have developed a variety of defense mechanisms to counteract the ROS-induced damages, such as increased accumulation of non-enzymatic and activities of enzymatic antioxidants, including glutathione, ascorbic acid, carotenoids, tocopherols, CAT (EC 1.11.1.6), ascorbate peroxidase (APX) (EC 1.11.1.11), guaiacol peroxidase (GPX) (EC 1.11.1.7) and SOD (EC 1.15.1.1) (*Rajput et al., 2021*).

Different phenolic compounds are well-known antioxidants that enhance oxidative stress tolerance in plant tissues (*Chen et al., 2019*). Though plants have a well-known mechanism to cope with stresses, their extent of defense is plant-specific and type stress-specific. Different ways are being used to enhance plant stress tolerance against different stresses. It has been reported that the application of practical, affordable, and cheap chemicals is found adequate to enhance plant tolerance to biotic and abiotic stress, including water stress (*Youssef et al., 2021*; *Akhter et al., 2023*; *Ramadan et al., 2022*).

Amongst these, ALA is also considered one of the novel substances, but few recent reports are available. Mainly, these studies are conducted in pots rather than in field conditions focusing on the salt, osmotic, and heavy metal stresses (*Terzi et al., 2018*; *Youssef et al., 2021*; *Rajput et al., 2021*; *Ramadan et al., 2022*), while the present study was conducted under field water deficit conditions. Moreover, earlier studies focused on earlier vegetative stages, either the attributes studied or, in the case of ALA exogenous application. Whereas in the present study, the ALA was applied as seed priming or as a foliar spray at the vegetative stage

Alpha lipoic acid (formula, $C_{18}H_{14}O_2S_2$, molar mass 206.33 g/mol) is a small molecule of disulfide, which acts as a coenzyme of $\alpha$-ketoglutarate dehydrogenase and pyruvate dehydrogenase in mitochondria. Reed first isolated it from the liver in 1951 and discovered it in 1973 (*Reed et al., 1951*). It is found in eukaryotic and bacterial organisms (*Terzi et al., 2018*). ALA and its reduced form, dihydrolipoic (DHLA), have powerful antioxidant potential which could reduce the ROS levels and free radicals, chelate metal ions, weaken oxidative stress, promote the endogenous antioxidant regeneration such as coenzyme Q10, glutathione, as well as its levels (*Liu et al., 2005*; *Alban et al., 2018*). Moreover, ALA is the only known antioxidant with lipo-soluble and water-soluble properties. Unlike liposoluble antioxidants, which only act on the cell membrane, and unlike water-soluble antioxidants, which only act on cytoplasm, ALA plays a dual protection role that can function simultaneously on the cell membrane as well as the cytoplasm (*Çelik & Ozkaya, 2002*).

Due to its two sulfhydryl moieties, it binds with metals and enables them to scavenge the free radicals responsible for its antioxidant capacity (*Fogacci et al., 2020*). Exogenous ALA application has been found to mitigate lipid peroxidation and regulate the osmotic potential and leaf photosynthetic performance under abiotic stresses (*Gorcek & Erdal, 2015*; *Sezgin et al., 2019*; *Elkelish et al., 2021*; *Youssef et al., 2021*; *Alomran et al., 2023*). Exogenous use of ALA increases the enzymatic antioxidant activities such as monodehydro ascorbate reductase (MDHAR), glutathione reductase (GR), GPX, CAT, and SOD under osmotic stress (*Rajput et al., 2021*; *Terzi et al., 2018*). Additionally, ALA has been found to be involved in restoring grain yield and quality attributes of water-stressed wheat plants (*Elkelish et al., 2021*). Under diverse environmental stress, ALA has been found helpful for the induction of stress tolerance mechanisms in several plant species by improving photosynthesis (*Turk et al., 2018*; *Terzi et al., 2018*; *Sezgin et al., 2019*).

Only a few reports regarding ALA's role in all the dating negative impacts of water deficit stress are available and only in the class. Moreover, regarding legumes, the role of exogenous ALA through different modes in the induction of water stress tolerance has not yet been reported/explored. It was hypothesized that exogenous application of ALA through different modes may ameliorate the water stress-induced adverse impacts on the seed yield of mungbean lines. The study's objectives were to determine the ameliorative impacts of deficit irrigation on the yield of two differentially water stress tolerant mungbean lines in relation to the plant water relations, photosynthetic pigments, lipid peroxidation, and oxidative defense mechanism. The study's objectives were also to select the most effective level of ALA, either applied as seed priming, foliar spray, or a

combination of both, for better seed yield of mungbean in field conditions under deficit irrigation that is still not found yet.

## MATERIALS AND METHODS

### Experimental conditions

The present experiment was conducted to assess the response of two high-yielding mung bean lines (16003 and 16004) to exogenously supplied ALA through different modes when grown under field water deficit conditions. Line 16003 is drought-sensitive, while line 16004 is moderately drought-tolerant (*Pulses Research Institute, Faisalabad, 2019*).
The whole experiment was conducted in the Adaptive Research Complex located in Sheikhupura, Punjab, Pakistan. The experiment was conducted in two consecutive years from September to December 2020 and 2021. However, the data presented for different attributes in this study is given only for a year due to similar findings. These selected mungbean lines are being cultivated by the farmers due to their high-yielding potential. The seeds of these lines were obtained from the lentil section of the Ayub Agriculture Research Institute (AARI). There were two regimes of irrigation, *i.e.*, normal irrigation (three irrigations during the growth period) and reduced irrigation (water stress) (only once). The experiment was arranged in a split-split plot design with three replicates for each treatment. The experimental area comprised two main plots corresponding to each irrigation regime (normal irrigation and reduced irrigation). Each main plot was divided into two subplots, each specific for the mungbean line. Each subplot was further divided into three subplots corresponding to the ALA mode of application (Foliar spray, seed priming, and foliar+priming). Each sub-sub plot contained nine furrows of 6 m in length with a furrow-to-furrow distance of 60 cm. A set of three furrows was specified for each specific level of ALA (0, 0.1, 0.15 mM of ALA). These doses of ALA as exogenous use were selected based on the available studies regarding its exogenous use (*Elrys et al., 2020*; *Yadav, Gupta & Seth, 2022*). For seed priming treatment, the seeds of both mungbean lines were primed separately with each specific level of ALA solution for 6 h before sowing. After soaking, the seeds were air-dried until the constant weight was achieved. Forty seeds were hand-sown in each row at a 15 cm distance. The sowing for soaked and non-soaked seeds was done at the same time. Before sowing, the soil was well prepared by ploughing when the soil was at the field capacity.

Thinning was done after 8 days of seed germination to maintain 30 cm plant-to-plant distance. During land preparation, the soil was supplied with an adequate amount of fertilizer [N (160 Kg/ha), P (80 Kg/ha), and K (50 Kg/ha)] as per the recommendations. Mungbean lines specified for foliar spray were sprayed with specific levels of ALA in the evening for the maximum absorption of the applied solution. Tween-20 (0.1%) was added to solutions prepared for each treatment prior to the foliar spray. The soil was covered with a polythene sheet before the foliar application, and then applied the ALA spray on each plant with the help of a spray bottle for the accuracy of results. Fifty milliliters of solution of ALA of each treatment were applied to each specific row, which had 20 plants in each row. The foliar spray to plants specified for dual treatment (seed priming+foliar) was also sprayed at the same time. After 2 weeks of foliar spray, five plants were harvested from

each treatment for the estimation of different growth, morphological, and biochemical attributes. At the same time, the yield attributes were recorded at maturity. The fresh leaf samples were collected after 2 weeks of application and stored at −80 °C to be used further for different biochemical analyses.

The growth parameters measured were the shoot fresh weight (SFW), root fresh weight (RFW), shoot dry weight (SDW), root dry weight (RDW), shoot length (SL), and roots length (RL). The biochemical analysis measured included the leaf photosynthetic pigments such as leaf chlorophyll *a* (Chl. *a*), chlorophyll *b* (Chl. *b*), total chlorophyll contents (T. Chl.) and chlorophyll *a/b* ratio (Chl. *a/b*), leaf proline content, glycine betaine (GB), total phenolics content (TPC), ascorbic acid (AsA) content, total flavonoids content (TFC), total soluble proteins (TSP), leaf hydrogen peroxide ($H_2O_2$) levels and leaf malondialdehyde contents (MDA) as well as the leaf antioxidants enzymes such as superoxide dismutase (SOD), peroxidase (POD) and catalase (CAT). The plant samples specified for fresh biomasses were placed in an electric oven at 70 °C for 48 h to get the dry weight after measuring the fresh masses of roots and shoots.

## Climatic conditions during experimentation

The averaged climatic conditions during the experimental period from September to December 2020 were as follows; photosynthetically active radiations (PAR) 972 μmolm-2s-1 maximum temperature 28.6 ± 1.2 °C, minimum temperature 14.75 ± 1.3 °C, relative humidity day 52.02 ± 1.9%, relative humidity night 76.97 ± 3.5% and 4.17 ± 2.62 mm average rainfall pattern.

## Soil texture and physico-chemical properties

The soil of the experimental site was sandy loam having available total N (0.73%) and organic matter (1.15%), and having P (8.6 ppm). The saturation percentage of the experimental soil was 34%. The average EC and pH of the soil was 2.53 ds.m-1 and 7.8, respectively. The soil solution has the soluble $CO_3^{2-}$ (traces), $SO_4^{-2}$ (1.98 meq L$^{-1}$), $HCO_3^{-}$ (4.93 meqL$^{-1}$), $Ca^{2++}Mg^{2+}$ (14.3 meq L$^{-1}$), Cl- (8.52 meq L$^{-1}$), Fe (0.041 meq L$^{-1}$), Na (2.98 meq L$^{-1}$) and SAR (0.086 meq L$^{-1}$). The soil's physical and chemical properties were assayed following *Dewis & Freitas (1970)*.

## Estimation of growth and morphological attributes

Carefully, one plant per replicate was uprooted in order to measure the fresh biomass of the root and shoot. Following washing, the root sample was blotted dry with paper to remove any remaining water. The fresh biomass of the shoot and root samples was then measured with an electric balance. Using a meter rod, the length of the shoot and root were measured simultaneously. The dry biomasses of the shoots and roots of these plants were then measured after being maintained at 65 °C for 72 h.

## Determination of leaf photosynthetic pigments

The contents of Leaf Chl. *a* and Chl. *b* were estimated by using (*Arnon, 1949*) approach. An extract of 0.5 g of fresh leaf was made in 10 ml of acetone (80%). For 5 min, the extract was centrifuged at 10,000 × *g*. At 480, 645 and 663 nm, the absorbance of the supernatant

was measured using a spectrophotometer. Using the following formulas, the contents of Chl. *a* and Chl. *b* were determined:

Chl. *b* (measured in mg g-1 FW) = [22.9 (OD 645) −4.68 (OD 663)] × V/1,000 × W Chl. *a* (measured in mg g-1 FW) = [12.7 (OD 663) −2.69 (OD 645)] × V/1,000 × W V = volume of the leaf extract (measured in ml)

W = fresh weight of the leaf tissue (measured in g)

However, for the estimation of leaf carotenoid contents, the formula given by *Kirk & Allen (1965)* was used.

Carotenoids (mg mL$^{-1}$) = A car/Em 100% × 100

A Car (carotenoid) = (OD 480) + 0.114 (OD 663) − 0.638 (OD 645)

Em (Emission) = Em 100% = 2,500.

## Determination of leaf osmolytes content

Using *Grieve & Grattan*'s *(1983)* methodology the GB content was determined. Homogenized the 0.5 g dry leaf material using 10 ml of distilled water and kept it over-night at 4 °C. Centrifuged the samples at 10,000 × *g* for 10 min. Then the 1 ml of the extract and 1 ml of 2 N sulfuric acid was mixed properly and 0.2 ml of potassium tri-iodide solution (KI$_3$) was added to it. After cooling well, the mixture for 90 min in an ice bath, 2.8 ml of distilled water and 5 ml of 1-2 dichloroethane added to the solution. When the two layers formed then collect the lower layer by removing the upper layer. We measured the absorbance of lower layer at 365 nm spectrophotometrically.

Leaf proline content was determined following the method recommended by *Bates, Waldren & Teare (1973)*. Fresh leaf 0.25 g thawed in 10 ml of 3% sulfosalicylic acid solution and filtered the mixture. Then, we added 2 ml of filtrate into the test tube and 2 ml of acid ninhydrin along with 2 ml of glacial acetic acid was added in it. For the preparation of acid ninhydrin: ninhydrin 1.26 g was mixed in in glacial acetic acid (30 ml) and 6 *M* ortho–phosphoric acid (20 ml). Then the resultant mixture was incubated at 100 °C for an hour. Cool down the solution and 2 ml toluene were added to the mixture and vortexed well. We took the upper layer and measured the absorbance at 520 nm using the spectrophotometer.

## Estimation of activities of enzymatic antioxidants and the contents of non-enzymatic antioxidants

### Estimation of CAT, POD and SOD activities

For the estimation of the activities of antioxidant enzymes, TSP, and free amino acids the fresh leaf material (0.5 g) was homogenized in 50 mM potassium phosphate buffer having pH 7.8. The homogenize was centrifuged at 20,000 × *g* at 4 °C and the supernatant was stored at −20 °C and later on used for the estimation of activities of antioxidant enzymes and TSP. The activities of CAT and POD were determined using the methods adopted by *Chance & Maehly (1955)*. For the estimation of CAT, the reaction mixture (3 ml) contained 50 mM phosphate buffer (pH 7.8), 59 mM H$_2$O$_2$, and 0.1 ml enzyme extract. The enzyme extract (100 μL) was mixed in the last to start the reaction. The change in absorbance was recorded at 240 nm for 120 s at intervals of 20 s. For the estimation of POD

activity, the reaction mixture was prepared using 0.1 ml enzyme extract, 40 mM $H_2O_2$, 20 mM guaiacol and 50 mM phosphate buffer (pH 7.8). The change in the absorbance was measured at 470 nm for 120 s at intervals of 20 s.

The *Giannopolitis & Ries (1977)* approach was used to measure the activity of SOD. A reaction mixture (1 ml) including 50 µM NBT (NBT solution prepared in formamide), 13 mM methionine, 1.3 µM riboflavin, 75 nM EDTA, and 50 mM phosphate buffer (pH 7.8) was prepared using 50 µL of the enzyme extract. The reaction mixture was then placed within an aluminum foil and exposed to a 20 W bulb for 15 min. Before the reaction mixture was exposed to the light source, riboflavin was added. Every time, a blank sample was made without any extract added. Using a spectrophotometer, the absorbance of the reaction mixture was determined at 560 nm.

### Estimation of leaf AsA content

According to the method given by *Mukherjee & Choudhuri (1983)* was used for the estimation of leaf AsA content. Fresh leaf material (0.25 g) was ground in 10 ml of 6% TCA solution. After centrifugation the supernatant (4 ml) was mixed with 2 ml of 2% dinitrophenyl hydrazine in prepared in 9N $H_2SO_4$. One drop of 10% thiourea was added to the solution. The thiourea solution was prepared by mixing the 2 g of thio-urea with 14 ml ethanol and 5 ml of distilled water. Incubated the mixture for 15 min in a water bath. After that, cooled the solution at room temperature and 5 ml of 80% $H_2SO_4$ added. The absorbance of the solution was read a using spectrophotometer at 530 nm.

### Estimation of leaf TPC and TFC

Following the approach outlined by *Julkunen-Tiitto (1985)* was used to measure the seedling TPC. Fresh leaf 0.25 g mixed with 5 ml of 80% acetone and then centrifuged at $10,000 \times g$ for 10 min. Then, in a microfuge tube collected the resultant supernatant and stored at 4 °C. After it, 0.1 ml supernatant mixed with 2 ml distilled water along with Folin-Ciocalteu's phenol reagent (1 ml) and shake gently. Then to the above mixture 5 ml of 20% $Na_2CO_3$ was added and the final volume was made up to 10 ml by adding distilled water. The absorbance of the triturate was recorded at 750 nm by using spectrophotometer. The leaf TFC was determined spectrophotometry according to *Karadeniz et al. (2005)*. Using a mortar and pestle, one gram of plant leaf material was taken and ground in 20 ml of 80% aqueous methanol. The filtrate was then obtained by filtering. Add 0.3 ml of 5% $NaNO_2$ and 3 ml of distilled water to the filtrate (0.5 ml). The solution was allowed to stand at room temperature after thoroughly mixing. Following that, triturate and 0.6 ml of 10% $AlCl_3$ were combined 2 ml of 1M NaOH was also added after 6 min. Using distilled water, the solution's volume was kept at 10 ml. The final solution's absorbance was measured spectrophotometrically at 510 nm

### Estimation of TSP

For the determination of TSP content, the earlier obtained phosphate buffer solution was used as used earlier for the estimation of antioxidant enzymes. Supernatant (0.1 ml) was reacted with 2 ml of Bradford reagent and the absorbance was measured at 595 nm following the method given by *Bradford (1976)*.

### Estimation of H$_2$O$_2$ and MDA contents

The levels of MDA represent the extent of lipid peroxidation (membrane damaging) under the conditions of stresses due to overly produced ROS. Using the approach outlined by *Cakmak & Horst (1991)*. Fresh leaf (0.25 g) was ground in 1% TCA (3 ml) at room temperature. At 5,000 × *g* centrifuged the extract for 15 min. Then 1 ml of supernatant was mixed with 4 ml of 0.5% TBA prepared in 20% TCA solution. Incubate the solution at 95 °C for 50 min and the samples were then cooled at room temperature. The absorbance of the supernatant was measure at 532 and 600 nm by using the spectrophotometer. For the estimation of H$_2$O$_2$ content the method proposed by *Velikova, Yordanov & Edreva (2000)* was used. The supernatant 0.5 ml was mixed with 1M KI, and incubated for 50 min at 95 °C, and absorbance was taken at 390 nm after cooling well.

### Estimation of yield attributes

Two plants per replicate were harvested at maturity stage for determination of different yield attributes such as number of seed per pod, number of pods per plant, hundred grain weight (100 GW) and total grain yield. The pods were collected manually from the plants and then dried in sunlight.

## Statistical analysis

The data was subjected to ANOVA for statistical analysis to find out the significant differences using CoStat CoHort software. In the studied attributes between treatments to determine the significant variations, the CoStat Computer Program (Monterey, CA, USA, PMB 320, Windows version 6.303) was used. The computer program XLSTAT (Addinsoft, Paris, France) was used to find out the correlations among the studied attributes. By using R-studio (Version 4.2.2) principal component analysis (Package ggplot2) among studied attributes and heat map (Package: "cluster","factoextra", gplots) to detect the correlation among studied parameters and treatments with parameters were constructed.

## RESULTS

### Changes in growth attributes and oxidative stress markers

The growth of mungbean lines was increased significantly when supplied with ALA levels (already screened in the pilot experiment) and when applied with different modes under water stress, presenting marked ameliorating ALA responses against water stress (Figs. 1 and 2). The SFW, SDW, RFW, RDW, RL, and SL significantly decreased with water stress imposition. Application of ALA through different modes showed a significant increase in the values of studied growth parameters in both mungbean lines. Both levels, *i.e.*, 0.1 and 0.15 mM of ALA, showed non-significant differences with each but significantly higher values compared to non-treated ones. However, the mode of application of ALA showed a significantly variable response. These growth attributes showed a significant increase in plants raised from seeds primed with different levels of ALA compared to non-primed ones. Seed priming with different levels of ALA improved the SFW, SDW, RFW, RDW, SL, and RL compared to plants grown with other application modes. However, the foliar application in combination with priming showed comparatively limited effects. SFW and

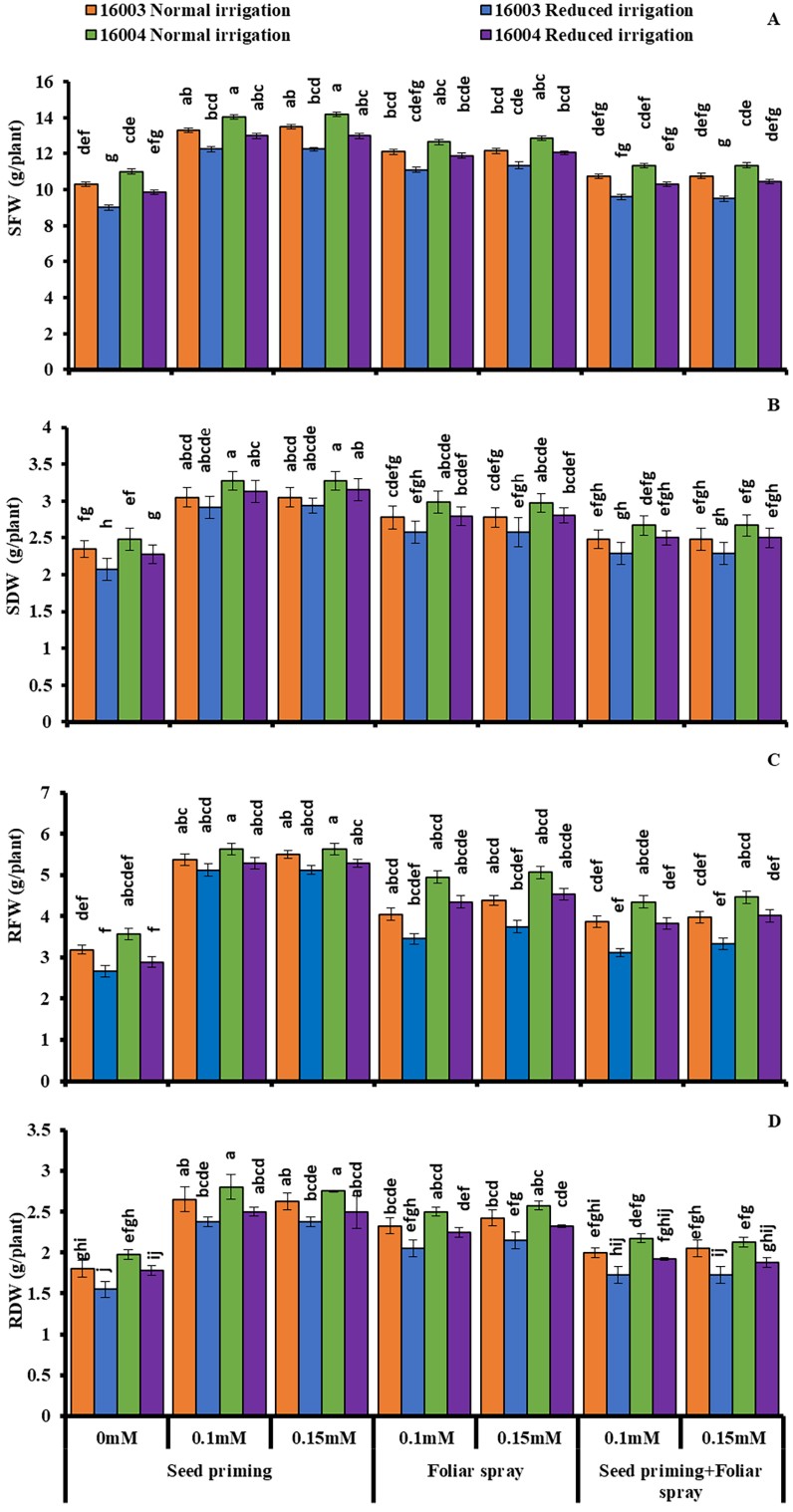

**Figure 1  SFW, SDW, RFW, RDW of differentially drought tolerant mungbean genotypes fertigated with different levels of ALA through different modes when grown under normal irrigation and reduced irrigation.** Means with the same letter are not significantly different from each other ($P > 0.05$ ANOVA followed by least significant difference (LSD). Error lines represent ± standard deviation of the mean.

SDW showed a significant increase under foliar application of both levels only under water stress conditions. Both levels of foliar application of ALA significantly increased the RDW in both mungbean lines. At the same time, RFW was significantly higher only in line (16004) under both water stress and non-stress conditions. Foliar application of 0.15 mM level significantly increased the SL and RL of both mungbean lines under water-stressed and nonstressed conditions than other treatments.

Imposing water stress markedly increased the membrane permeability $H_2O_2$ levels and MDA content in both mungbean lines compared with plants grown under nonstressed conditions. Moreover, different ALA concentrations through different modes (0.1, 0.15 mM) when caused marked decreases in membrane permeability $H_2O_2$ levels and MDA content. However, the ALA application mode response was observed to be significantly variable. For different modes of application as well as for different treatments, the decrease was different. Compared to other applications, the priming mode showed higher values (Fig. 2).

## Changes in leaf photosynthetic pigments

The plants raised under water stress conditions showed a significant decrease in leaf Chl. *a*, Chl. *b* of both mungbean lines as compared to plants grown under non-stress conditions. Line 16003 showed a significantly more significant decrease than line 16004. When applied through different modes, Exogenous application of ALA significantly increased both Chl. *a* and Chl. *b* contents under nonstressed and water-stressed conditions, but this increase was more with seed priming than other treatments. This behavior was noted for both ALA levels. However, foliar application of ALA showed a significant increase only in Chl. *b*. Chl. *The a/b* ratio and T. Chl. were also recorded to decline under water stress. However, when applied as a seed priming agent or through other modes, ALA significantly increased both Chl. *a/b* and T. Chl. under both nonstressed and water-stressed conditions in both mungbean lines, but more improvement was found due to seed priming than other treatments (Fig. 3).

## Changes in osmolytes and non-enzymatic antioxidants

The changes in the levels of osmoprotectants such as proline and GB of both lines of mungbean in response to exogenous treatment of different concentrations (0.1, 0.15 mM) of (ALA) and under water stress are presented in Fig. 4. Imposition of water stress markedly increased the proline and GB in both mungbean lines compared to plants grown under nonstressed conditions. Moreover, different ALA concentrations (0.1, 0.15 mM) when applied as seed priming agent or as foliar spray caused marked further increases in the studied proline content of both mungbean lines compared to their corresponding untreated ones. Similar results were found for GB, but only when ALA was applied as a priming agent, and foliar application was found effective only for nonstressed plants of both lines. The combined application of priming and foliar levels showed no significant increase in GB and proline in both lines and under nonstressed and water-stressed conditions. Applying ALA (0.1, 0.15 mM) as priming and foliar spray significantly increased the GB contents in nonstressed plants of line 16004.

Peerj

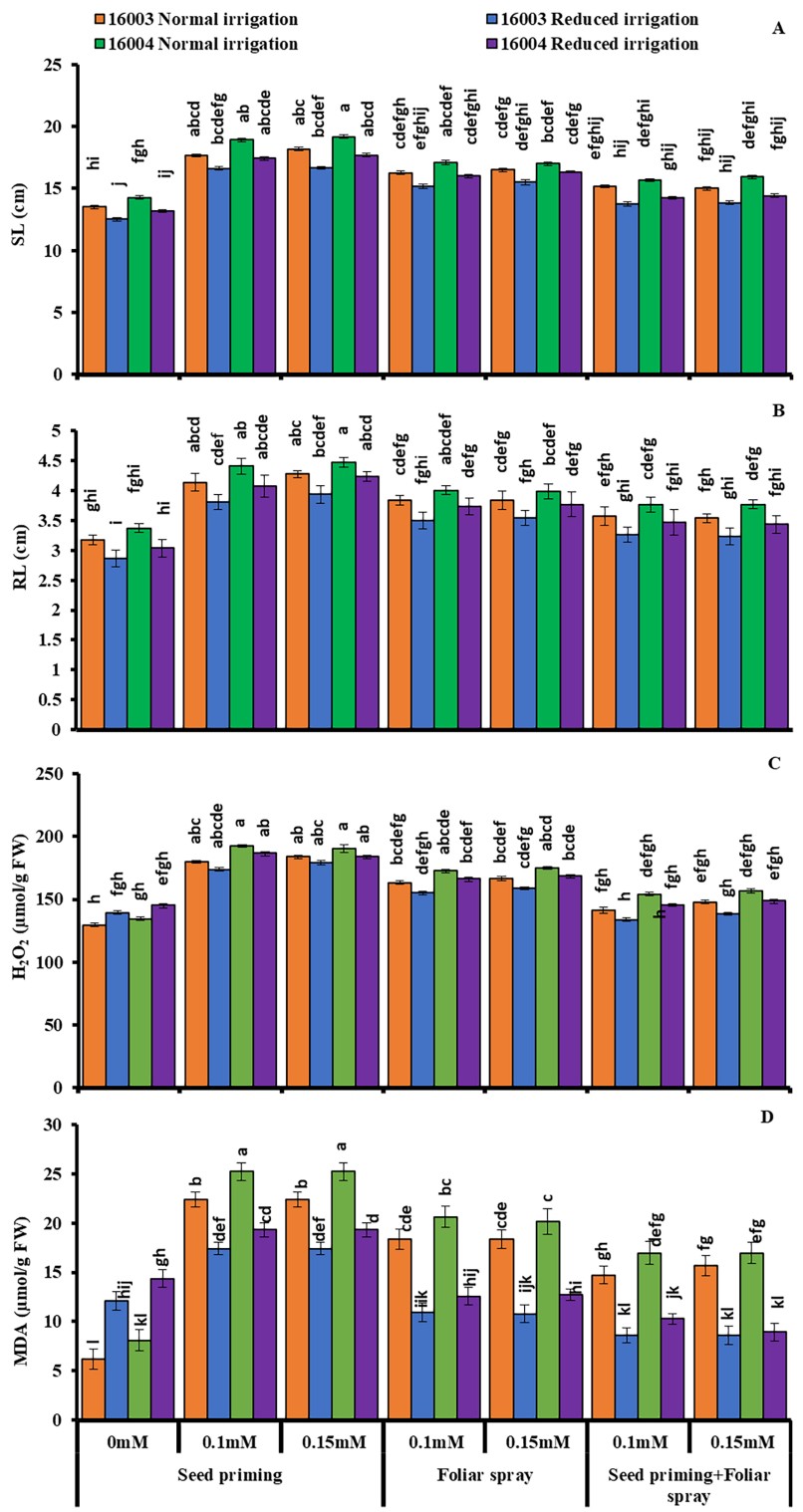

**Figure 2 SL, RL, H₂O₂, MDA of differentially drought tolerant mungbean genotypes fertigated with different levels of ALA through different modes when grown under normal irrigation and reduced irrigation.** Means with the same letter are not significantly different from each other ($P > 0.05$ ANOVA followed by least significant difference (LSD). Error lines represent ± standard deviation of the mean.

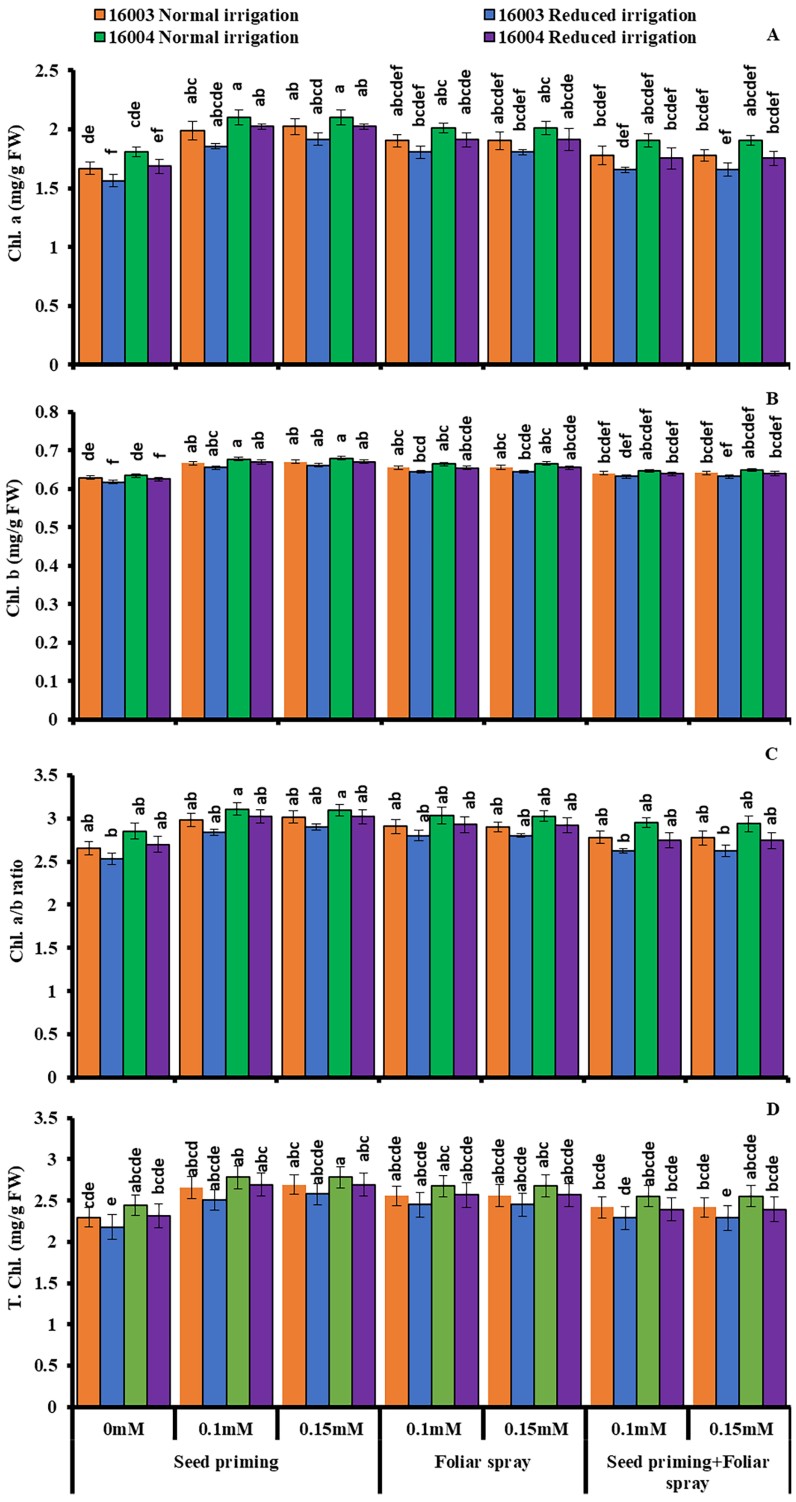

**Figure 3 Chl. *a*, Chl. *b*, Chl. *a/b*, T. Chl of differentially drought tolerant mungbean genotypes fertigated with different levels of ALA through different modes when grown under normal irrigation and reduced irrigation.** Means with the same letter are not significantly different from each other ($P > 0.05$ ANOVA followed by least significant difference (LSD). Error lines represent ± standard deviation of the mean.

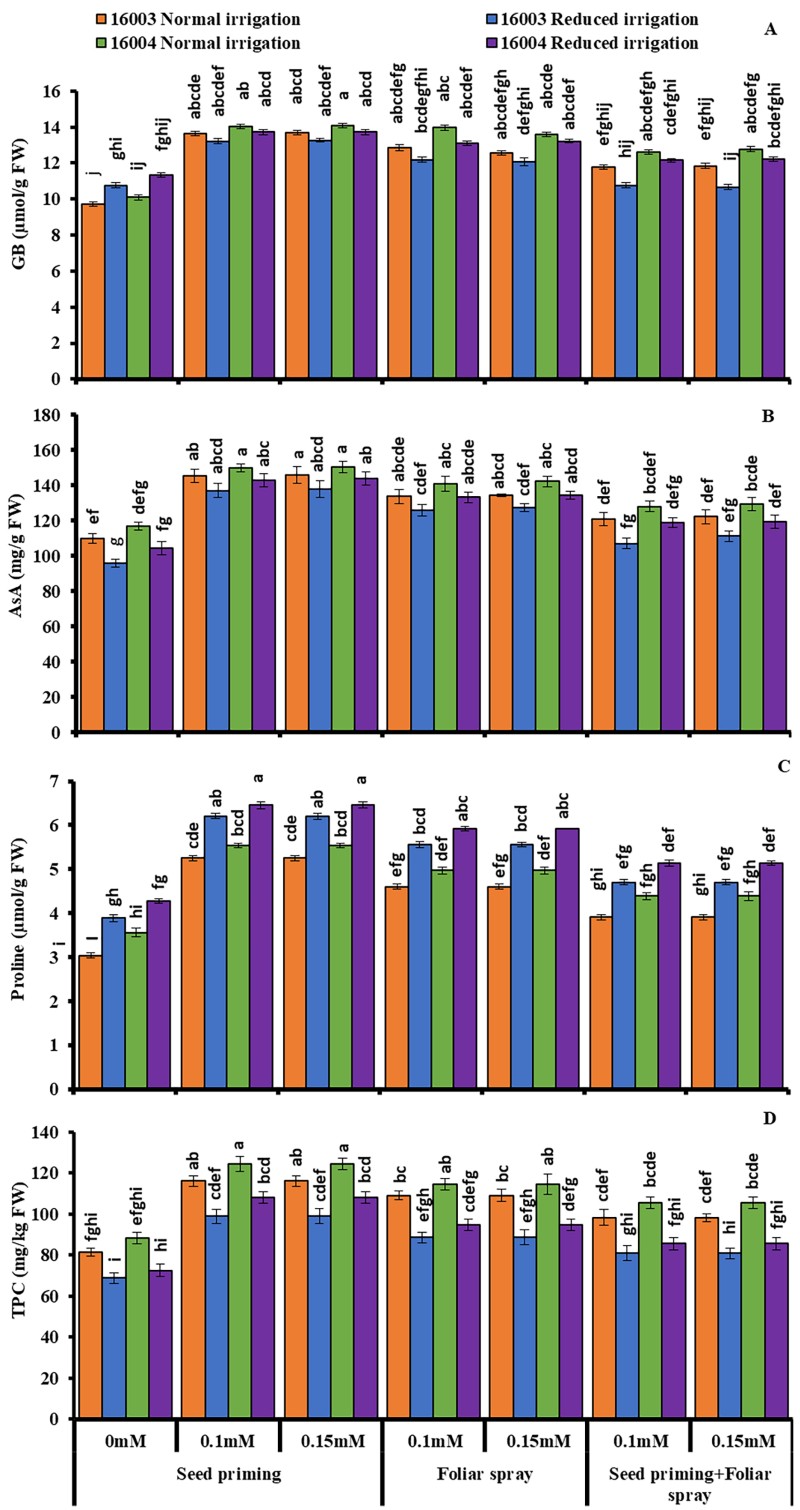

**Figure 4 Glycine betaine, AsA, proline, TPC of differentially drought tolerant mungbean genotypes fertigated with different levels of ALA through different modes when grown under normal irrigation and reduced irrigation.** Means with the same letter are not significantly different from each other (P > 0.05 ANOVA followed by least significant difference (LSD). Error lines represent ± standard deviation of the mean.

Imposition of water stress caused a significant decrease in leaf AsA contents of both mungbean lines relative to the nonstressed plants, as shown in (Fig. 4). Exogenous application of ALA as seed priming or as a foliar treatment with different concentrations (0.1, 0.15 mM) improved the AsA contents of both mungbean lines under nonstressed and water-stressed conditions compared to untreated plants. ALA application as pre-sowing seed treatment was found to be a more practical approach to increasing the leaf AsA contents in both mungbean lines when grown under non-stress and water stress conditions as compared with foliar application of ALA, which showed a significant increase in ascorbic acid content but only in nonstressed plants of line 16003.

Subjecting mungbean plants to water deficit, stress caused a significant decrease in TPC of line 16003 relative to the control plant (Fig. 4). Exogenous application of ALA through different modes with different concentrations (0.1, 0.15 mM) caused a significant increase in leaf TPC contents in both lines of mungbean as compared with their corresponding untreated control plants. Both levels were equally effective as priming agents or when applied at the foliar level, as they caused significant increases in TPC under both nonstressed and water-stressed plants of mungbean lines. However, the combined application of any level of ALA as seed priming or as foliar spray did not show any significant change in TPC contents under nonstressed and water-stressed conditions.

## TSP and TFC contents

Under water stress conditions, mungbean plants showed a significantly lower TSP and TFC content under water stress conditions than plants grown under non-stress conditions. Exogenous application of ALA with different concentrations (0.1, 0.15 mM) caused significant increases in TSP and TFC content in both lines of mungbean as compared with their corresponding untreated controls (Fig. 5). However, the extent of increase in TSP and TFC content was different in different mode of applications. TSP and TFC content showed a significant increase in plants raised from seeds primed with 0.1 and 0.15 mM levels of ALA compared to non-treated ones. The other modes of application, such as foliar application or foliar plus priming application of ALA, showed no significant increase from corresponding non-treated ones.

## Changes in enzymatic antioxidants

Under water stress conditions, mungbean plants showed a highly significant decrease in CAT, SOD, and POD activities as compared to plants grown under non-stress conditions. Exogenous application of ALA with different concentrations (0.1, 0.15 mM) caused significant increases in enzymatic antioxidant activity in both lines of mungbean as compared with their corresponding untreated controls (Fig. 6.). However, the extent of increase in antioxidants was different in different mode of applications. All these studied antioxidant enzymes showed a significant increase in plants raised from seeds primed with 0.1 and 0.15 mM levels of ALA compared to non-treated ones. The other modes of application, such as foliar application or foliar plus priming application of ALA, showed no significant increase from corresponding non-treated ones.

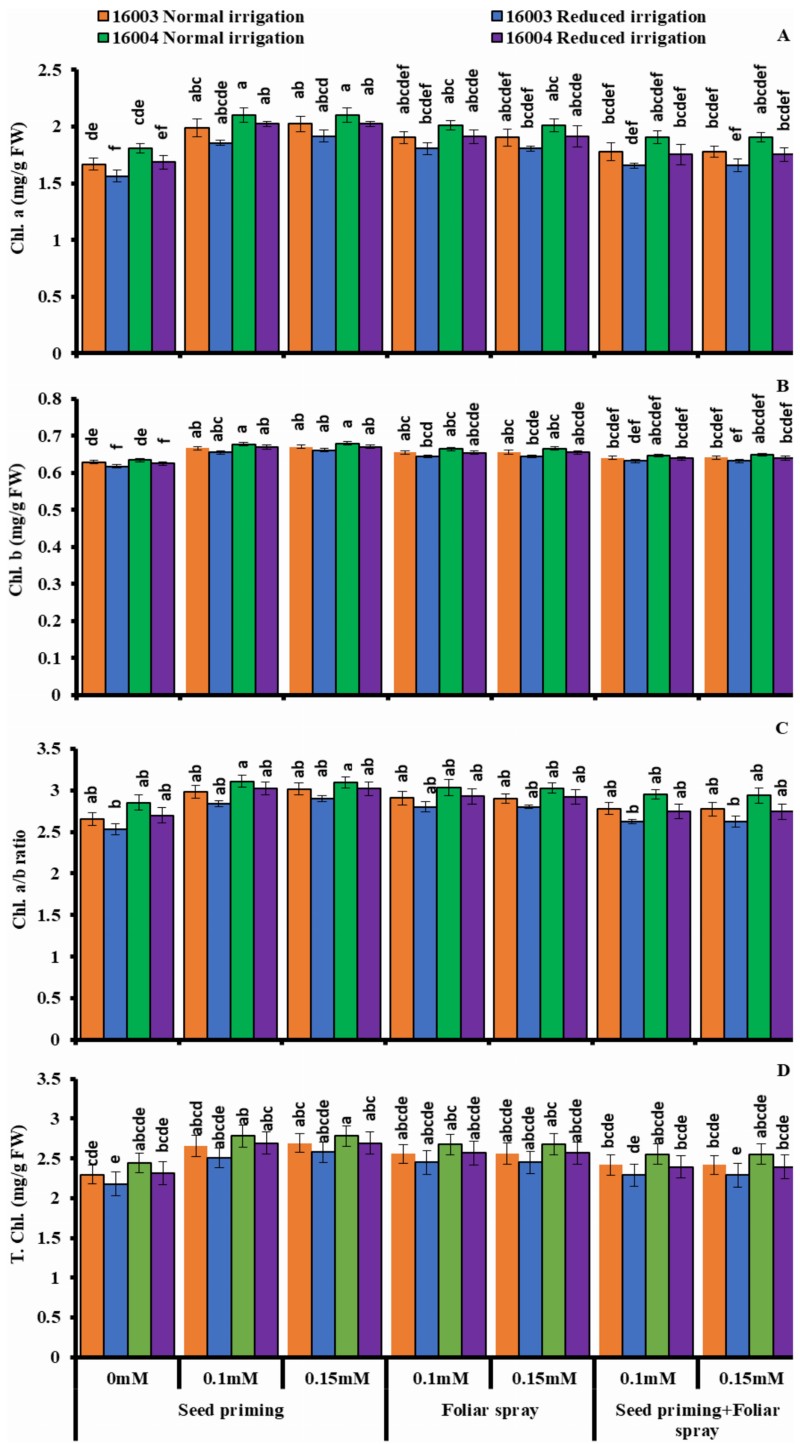

**Figure 5  TSP, TFC of differentially drought tolerant mungbean genotypes fertigated with different levels of ALA through different modes when grown under normal irrigation and reduced irrigation.** Means with the same letter are not significantly different from each other ($P > 0.05$ ANOVA followed by least significant difference (LSD). Error lines represent ± standard deviation of the mean.

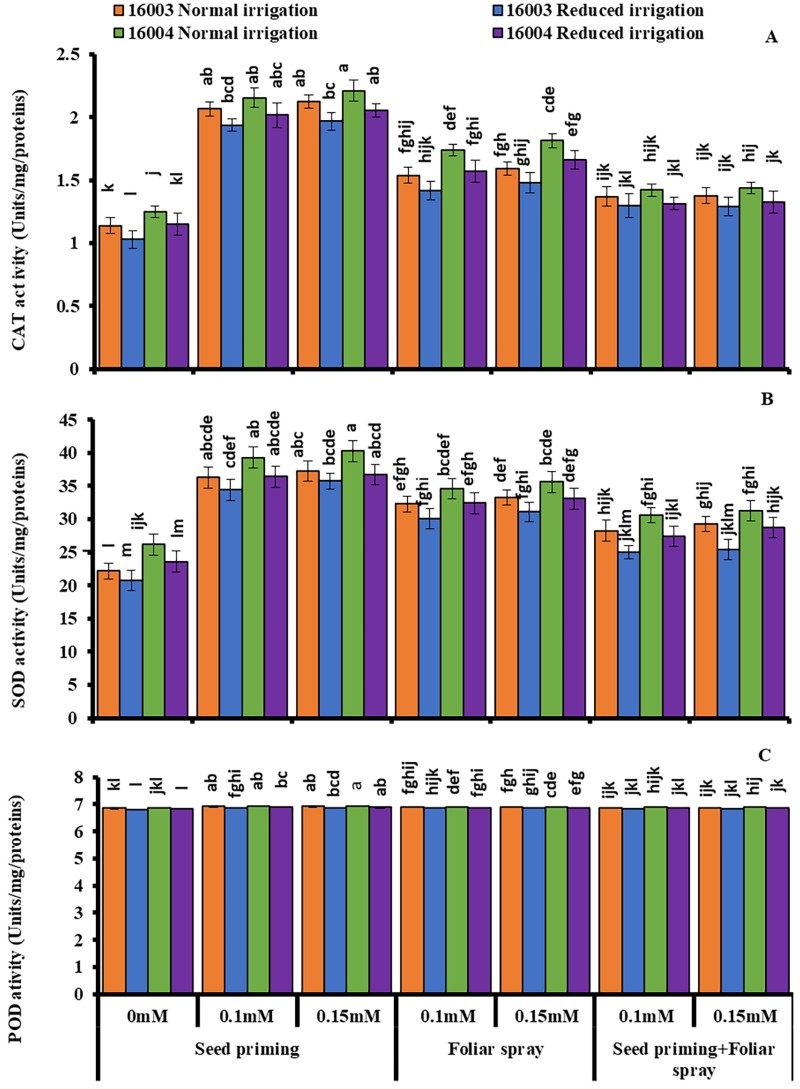

**Figure 6 Activities of CAT, SOD and POD of differentially drought tolerant mungbean genotypes fertigated with different levels of ALA through different modes when grown under normal irrigation and reduced irrigation.** Means with the same letter are not significantly different from each other ($P > 0.05$ ANOVA followed by least significant difference (LSD). Error lines represent ± standard deviation of the mean.

## Changes in yield attributes

Data presented in Fig. 7 shows seed yield and yield attributes such as the number of seeds per pod, the number of pods per plant, 100 GW, and TGY of two mungbean lines decreased markedly by imposing water stress compared to control plants. However, exogenous application of both levels of ALA (0.1, 0.15 mM) applied either by seed priming or foliar spray caused a significant increase in all parameters of yield components under nonstressed and under water-stressed conditions. On the other hand, combined application (priming+foliar) of any level of ALA did not show any significant change in this regard. Data show the superiority of line 16004 over line 16003 in yield and yield components.

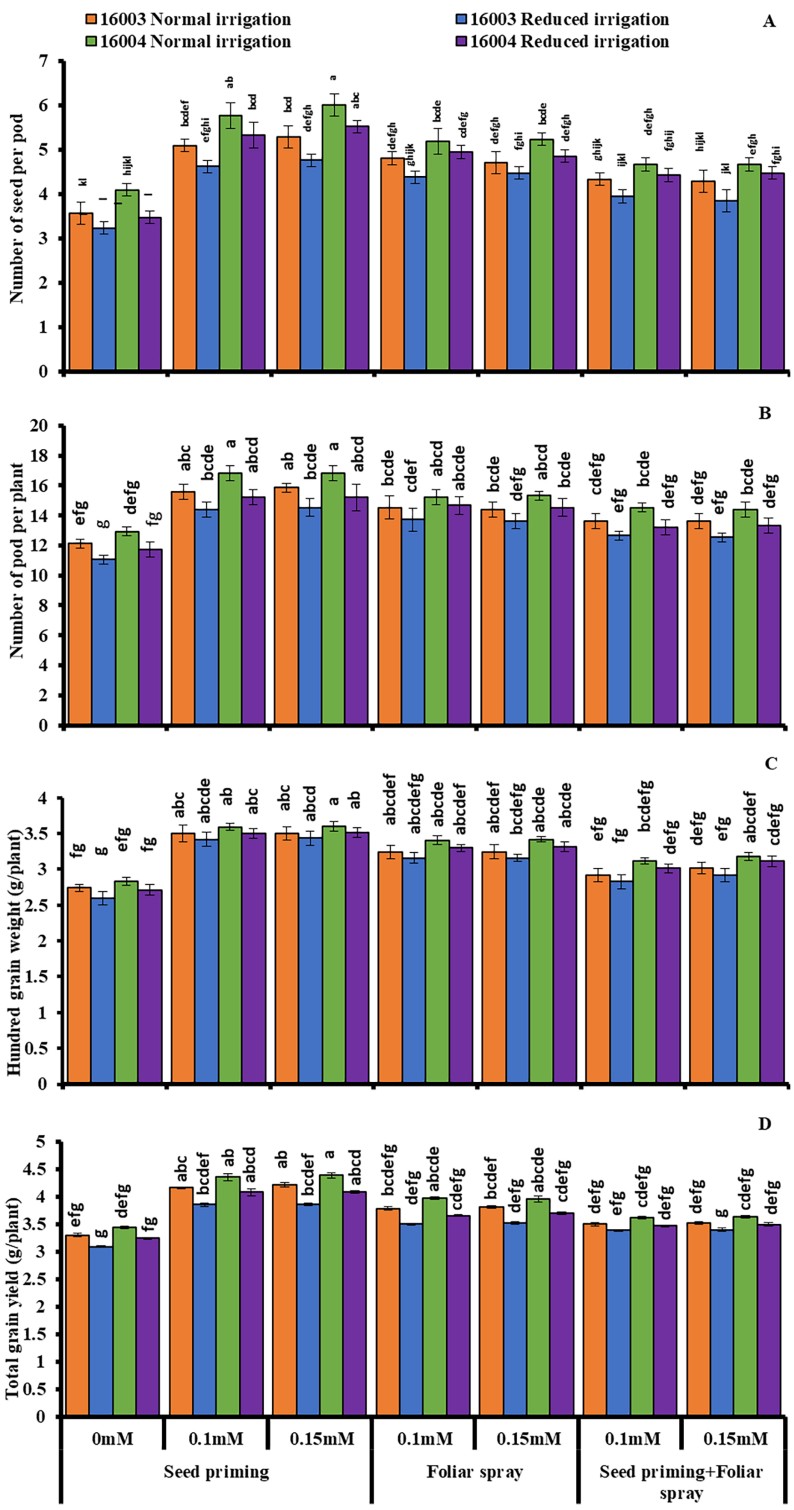

**Figure 7 Number of seeds per pod, number of pods per plant, 100 GW, TGY of differentially drought tolerant mungbean genotypes fertigated with different levels of ALA through different modes when grown under normal irrigation and reduced irrigation.** Means with the same letter are not significantly different from each other ($P > 0.05$ ANOVA followed by least significant difference (LSD). Error lines represent ± standard deviation of the mean.

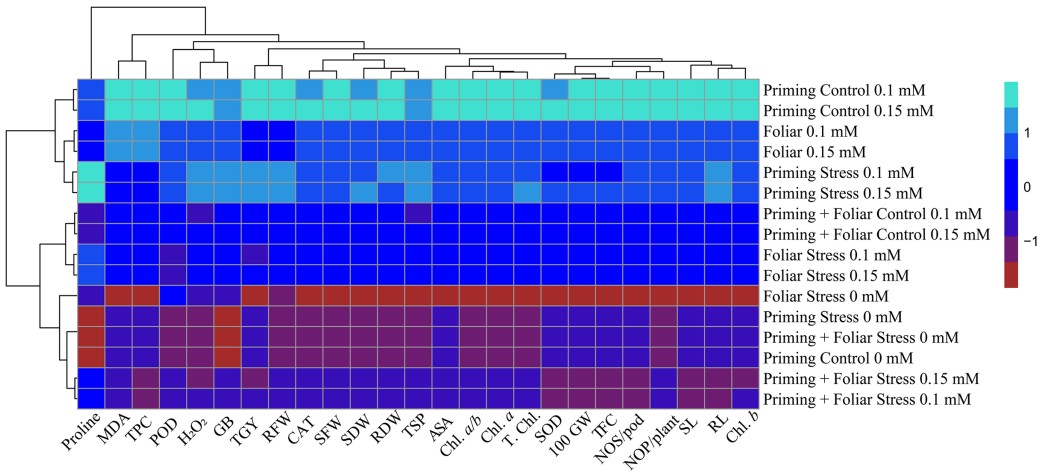

**Figure 8 Heatmap histogram correlation between different studied attributes of mung bean line (16003) fertigated with different levels of lipoic acid.** Priming ((Control 0 mM (1), 0.1 mM (2), 0.15 mM (3), Stress 0 mM (4), 0.1 mM (5), 0.15 mM (6)), Foliar (Control 0.1 mM (7), 0,15 mM (8), Stress (0 mM (9), 0.1 mM (10), 0.15 mM (11)) and Priming+ Foliar (Control 0.1 mM (12), 0.15 mM (13), Stress 0 mM (14), 0.1 mM (15), 0.15 mM (16)) when grown under normal irrigation and reduced irrigation.

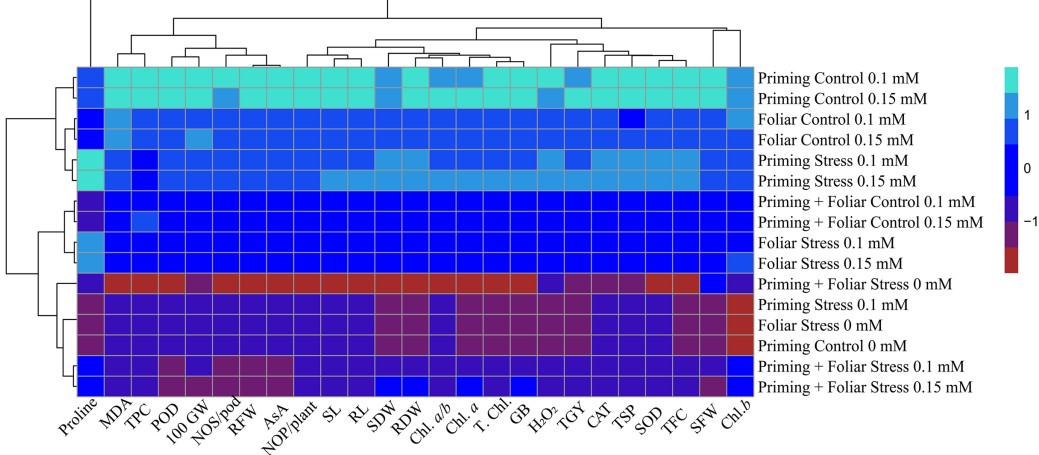

**Figure 9 Heatmap histogram correlation between different studied attributes of mung bean line (16004) fertigated with different levels of lipoic acid.** Priming ((Control 0 mM (1), 0.1 mM (2), 0.15 mM (3), Stress 0 mM (4), 0.1 mM (5), 0.15 mM (6)), Foliar (Control 0.1 mM (7), 0,15 mM (8), Stress (0 mM (9), 0.1 mM (10), 0.15 mM (11)) and Priming+Foliar (Control 0.1 mM (12), 0.15 mM (13), Stress 0 mM (14), 0.1 mM (15), 0.15 mM (16)) when grown under normal irrigation and reduced irrigation.

## Heat map

Data presented in Figs. 8 and 9 regarding heatmap histogram correlation shows not only genotypic responses of mungbean lines to water deficit stress but also corresponds well to the responses of lines to exogenously applied different levels of ALA through different modes. The intensity of the color of the squares in columns and lines describes the intensity of positive and negative relationships among attributes. The x-axis categories in

the case of genotype line 16003 4, 9, and 14 show strong negative correlation (dark red colors) with maximum studied attributes except for $H_2O_2$, MDA, and proline, where a positive correlation was found without treatments, and others showed less negative correlation. However, in the case of genotype line 16004, the intensity of negative impacts was less, as present in brown color squares against x-axis attributes with categories 4, 14, and 9 at the y-axis (Fig. 9). It shows a better performance of line 16004 in comparisons to line 16003. Heatmap histogram based on intensities of colors (blue color) shows that a strong positive correlation can be observed of categories 2 (0.1), 3 (0.15), 5 (0.1), and 6 (0.15 mM) as seed priming treatment under non-stress and stress conditions respectively with studied attributes of both mungbean lines but comparatively more positive responses were found in genotype line 16004. Categories seven and eight correspond to foliar treatments of 0.1 and 0.15 mM ALA, which also showed positive influence but were less than the pre-sowing seed treatment. Foliar spray of ALA 0.1, 0.15 mM under stress, and the combined treatments were not found so positively effective in both mungbean lines and showed an intermediate response as shown by light green squares in column and lines against the attributes x-axis.

## DISCUSSION

One of the environmental stresses responsible for the decrease in plant growth and productivity is a deficit of water for irrigation. Under deficit irrigation, plants undergo numerous metabolic modifications for their survival. The present investigation recorded a significant decrease in growth and yield in two mungbean lines when grown under water stress. The results are in harmony with the studies of Hossain et al. (2020), who reported adverse impacts of deficit irrigation on growth of canola, wheat, quinoa, and moringa plants that was associated with increased lipid peroxidation due to overly produced ROS, reduced cell enlargement due to disturbed water relation (Shahid et al., 2022). Water stress-induced negative impact on plant water relations causes osmotic stress that directly affects the imbibition process necessary for cell expansion, leading to cell division responsible for continuity of growth (Shehzad et al., 2022; Wakchaure et al., 2023).

Moreover, water deficit conditions directly suppress the development of optimal leaf area, which in turn causes a decrease in photosynthesis and consequently the growth and yield (Seleiman et al., 2021), similar to the present investigation. However, these reductions in growth and yield attributes of mungbean plants due to water stress were less than those supplied with ALA through different modes and comparatively better in plants grown from seeds treated with ALA than foliar or combined application. It shows a clear role of ALA in plant stress tolerance due to involvement in different physiological and biochemical activities. Earlier studies revealed the promotive role of ALA on growth attributes of plants grown under stress (Youssef et al., 2021) who reported that ALA acts as a growth regulator and reported as a protector against abiotic stress (Ramadan et al., 2022). Moreover, it was reported earlier that exogenous application of ALA promoted root system in canola seedlings grown under stress (Javeed et al., 2021) that was found helpful in promoting the plant shoot system and growth attributes associated with increased uptake of nutrients and water, leading to modulation in physiological process such as (Ramadan

*et al., 2022*) photosynthesis and increased antioxidant activities (*Youssef et al., 2021*). In the present experiment, it was found that ALA acts as a potential modulator of plant growth under stress in a mode of application-dependent manner, where seed priming mode of application was found to be superior as compared to the foliar or foliar plus priming treatment, for better growth and yield.

Moreover, ALA improved yield has already been reported earlier (*Xiao et al., 2018*; *Elkelish et al., 2021*), alpha lipoic acid-induced application as a potential antioxidant reported to accelerate the growth of plants associated with cell enlargement, cell division due to maintenance in membrane integrity and reduced ion linkage (*Muscolo et al., 2014*; *Hassan et al., 2021*). In plants, photosynthetic pigments such as chlorophylls are indispensable to maintain optimum photosynthetic efficiency by harvesting sunlight (*Tabassum et al., 2016*). Under water stress conditions, leaf chlorophylls contents disturb seriously but plant species-specific (*Zargar et al., 2017*; *Shin et al., 2021*) and water stress intensity specific (*Zargar et al., 2017*; *Wang et al., 2016*; *Hussain et al., 2018*; *Tahir et al., 2019*; *Ma et al., 2022*) as well as in mungbean cultivars (*Uddin, Ullah & Nafees, 2021*) that corresponds well to present findings. Its results in stomatal closure lead to perturbation in gas exchange attributes associated with reduced leaf area (*Zargar et al., 2017*; *Liang et al., 2020*). Another factor responsible for the decrease in photosynthetic pigments might be the reduced transcription of the cab gene family responsible for the biosynthesis of chlorophyll molecules due to decreased *de novo* synthesis of said pigments and further destruction of the pigment-protein complexes (*Paim et al., 2020*) chloroplast lipids also contribute in declining the Chl. *a*, Chl. *b* and carotenoid contents (*Liang et al., 2020*; *Uddin, Ullah & Nafees, 2021*).

In the present study, exposure of mungbean plants to water deficit conditions noticeably altered the leaf chlorophyll contents. It was found that the Chl. *a*, Chl. *b*, Chl. *a/b* ratio and T. Chl. contents were significantly decreased in the water-stressed mungbean plants compared to the well-watered counterparts, which could be attributed to the disturbed activity of chlorophyllase (*Ali et al., 2018*) as well as oxidative damages (*Ali et al., 2018*; *Hassan, Ebeed & Aljaaran, 2020*). However, on the other hand, plants exogenously supplied with ALA showed less reduction in photosynthetic pigments, including Chl. *a* and Chl. *b*, and Chl. *a/b* ratio and T. Chl. under water-stressed conditions that show the antistress roles of ALA.

Similar ameliorations in stress-induced damages to pigments by exogenously applied ALA have previously been reported, where ALA-induced improvement in Chl. *a*, Chl. *b* and T. Chl. contents under stressful conditions associated with reduced lipid peroxidation by stimulating the antioxidant systems in wheat and maize (*Sezgin et al., 2019*; *Elkelish et al., 2021*; *Ramadan et al., 2022*). This indicates the species-specific role of ALA. Moreover, *Youssef et al. (2021)* reported that ALA application maintains the ultra-structure of chloroplasts for preserving the chlorophyll, proving the promoting role of ALA in photosynthesis under water stress.

Due to sulfur-containing biomolecules, ALA is considered a very effective antioxidant to protect plants when grown under abiotic stresses (*Gorcek & Erdal, 2015*). It was found that exogenous application of ALA stimulates not only the PS-II but also regulates the gene

expression of Rubisco and PEP carboxylase like certain carbon fixation enzymes in maize seedlings when grown under water stress conditions with a concurrent down-regulation in the chlorophyllase gene (Chlase) and an increase in N uptake, essential for the biosynthesis of chlorophyll (*Sezgin et al., 2019*; *Sadak et al., 2020*). These findings can be correlated well with present results where the exogenous application of ALA in any mode effectively reduced the adverse impact of water stress on leaf photosynthetic pigments.

Several evidences demonstrate that increased uptake of water and osmotic potential under osmotic stress (salinity or water stress) in plants is usually associated with accumulation of considerable concentrations of variety of organic molecules that improve plant water relations by performing the role of osmo-regulators and avoid disintegration of protein working as crops (*Majumdar et al., 2016*; *Alnusairi et al., 2021*; *Ibrahim, Ibrahim & Abd El-Gawad, 2021*; *Nahhas et al., 2021*; *Shao et al., 2021*; *Irshad et al., 2022*). Accumulation of proline maintains the integrity of the membrane, decreasing lipid oxidation through scavenging free radicals (*Shinde et al., 2016*) and acts as a signaling compound by regulating the function of mitochondria and controls the proliferation of cells by activating particular antistress genes (*Meena et al., 2019*). In the present study, imposition of water stress caused a marked increase in leaf proline and GB accumulation in both mungbean lines, showing their osmoregulation potential.

In the current study, exogenously applied ALA further increased the accumulation of proline and GB in mungbean plants of both lines under water stress, that are in agreement with former reports on different crop plants (*Mohammad-khani & Heidari, 2008*; *Terzi et al., 2018*; *Sezgin et al., 2019*; *Elkelish et al., 2021*; *Youssef et al., 2021*) where they reported the role of ALA in plant water relation through osmotic adjustment. In contrast to our findings, a reduction in leaf proline content was recorded in ALA-treated stressed wheat plants (*Ramadan et al., 2022*).

ROS production, known as oxidative stress, is a common phenomenon under water stress. The ROS accumulation, such as $H_2O_2$, leads to significant lipid peroxidation (*Ali & Ashraf, 2011*; *Shehzad et al., 2022*). In the present study, water stress significantly increased leaf $H_2O_2$ content, but the exogenously-applied ALA in any mode significantly decreased its level, which is in agreement with previous studies (*Youssef et al., 2021*; *Ramadan et al., 2022*) where ALA application reduced the lipid peroxidation by playing an antistress role. However, the interaction between increased ROS levels and ALA under water stress remains to be discovered, as limited literature is available regarding this treatment. To scavenge the overly produced ROS, plants have developed antioxidant defense mechanism compounds responsive to the reduction of lipid peroxidation but strongly cultivar/species specific and are measured in terms of MDA content (*Ali & Ashraf, 2011*; *Ali et al., 2016*; *Shehzad et al., 2022*). In the present study, MDA content increased significantly in both mungbean lines under water stress. However, ALA application lowered the MDA levels, as reported in stressed sorghum (*Youssef et al., 2021*) and wheat (*Ramadan et al., 2022*). Our results revealed that water stress induced substantial damage to the cell membrane, as depicted by increased $H_2O_2$ levels in both mungbeans. However, ALA-induced decline in MDA and $H_2O_2$ indicates a better antioxidant system in the treated plants, as reported in wheat and yeast (*Gorcek & Erdal, 2015*). Exogenous application of ALA restored the

decreased levels of other antioxidants. It reduced the damaging impact of oxidative stress *in vivo* under different physiological conditions (*Moini, Packer & Saris, 2002*), proving its role in protecting from stress-induced lipid peroxidation (*Terzi et al., 2018*). ALA recycles oxidized radical scavengers such as AsA and GSH (*Navari-Izzo, Quartacci & Sgherri, 2002*). Thus, it is a potential candidate for antioxidants, which can mitigate the oxidative damage induced by abiotic stresses (*Terzi et al., 2018*). This positive role of ALA in the improvement of the antioxidative defense mechanism is associated with improved osmotic adjustment and water relation and less damage to photosynthetic pigments as a result of better growth under water stress.

Under stresses, AsA plays crucial role in plants that is plant species-specific (*Naz et al., 2022*; *Alizadeh, Kumleh & Rezadoost, 2023*). In the present study, water stress significantly decreased the leaf AsA contents; however, exogenously-applied ALA in any mode improved its content in both lines of mungbean, the best being more in the case of seed priming. Previously similar rise in AsA content was noted in ALA-treated plants of wheat under osmotic stress (*Ramadan et al., 2022*), hence providing a close link between the attenuation of the oxidative damage and exogenous application of ALA mediated by non-enzymatic antioxidants.

Moreover, being potent antioxidants, TPC and TFC accumulation, along with other secondary metabolites, mitigates oxidative damage (*Wang et al., 2016*; *Ahmad et al., 2019*; *Li et al., 2019*; *Yadav et al., 2021*). In the present study, water stress caused a significant reduction in TPC and TFC in both mungbean lines that are similar as reported earlier (*Krol, Amarowicz & Weidner, 2014*; *Ali et al., 2018*; *Kumar et al., 2023*), where significant reductions in TPC and TFC where the phenolic compounds are being involved in plant tolerance to stresses (*Ferreyra, Rius & Casati, 2012*; *Saeed et al., 2023*). In the present study, exogenously applied ALA significantly reduced the adverse impacts of water stress on leaf TPC in both lines of mungbean. Similar findings were recorded in wheat under water stress (*Elkelish et al., 2021*) and osmotic stress (*Ramadan et al., 2022*). This improvement in polyphenolic content is positively associated with their better growth, better photosynthesis pigmentation, and reduced lipid peroxidation, which shows the stress tolerance role of exogenously applied ALA in both mungbean lines when applied through different modes, being better as a seed treatment.

In the present study, activities of antioxidant enzymes such as CAT, SOD, and POD decreased under stress in both mungbean lines, but reports depict that up or down-regulation activities of antioxidant enzymes or plant genotype-specific and stress type-specific (*Wang et al., 2012*; *Rai, Morales & Aphalo, 2021*; *Youssef et al., 2021*; *Bashir et al., 2023*; *Urmi et al., 2023*). In the present study, exogenously applied different levels of ALA through different modes were found effective in altering the CAT, SOD, and POD activities in favor of a better antioxidative defense mechanism. In an earlier study, it was reported that exogenous application of ALA elevated the antioxidant enzymatic activities such as SOD, CAT, MDHAR, GPX and GR under osmotic stress than to seedlings of maize not exposed to ALA (*Terzi et al., 2018*) and in water-stressed *Triticum aestivum* plants (*Elkelish et al., 2021*) that was associated with their stress tolerance in term of better growth due to better oxidative stress performance. From the present findings, as well as the

previous studies, it is confined that ALA applied improved antioxidative mechanism by upregulation of enzyme activities is specific to the type of stress, ALA level, and plant species (*Huang et al., 2019*; *Elkelish et al., 2021*; *Hassan et al., 2021*; *Youssef et al., 2021*).

It is further advocated that ALA contributes to instigating cellular redox management and ROS scavenging activities (*Turk et al., 2018*; *Terzi et al., 2018*; *Perveen et al., 2019*). Further investigations are needed to explore the mechanism of ALA in signaling pathways in modifying antioxidative enzymes at the organelles level in response to water stress conditions.

From the findings of the present study, it can be concluded that water stress tolerance induction in mungbean line by exogenous use of ALA through different modes in terms of better growth and yield is associated with better maintenance of plant water relations through osmotic adjustment, improvement in antioxidative defense mechanism with reduced lipid peroxidation and reduced membrane leakage, maintenance of better photosynthetic pigments in relation with reduced lipid peroxidation and better antioxidative defense mechanism. However, variable-limited reports in the literature pointed out its species and dose-dependent behavior. Therefore, similar studies on other crops are needed in the future to determine their exact stress tolerance mechanism. Additionally, there needs to be more information about pathway modulation associated with ALA-induced stress tolerance. Therefore, such studies should be the focus of the future. Thus, comprehensive molecular-level studies concerning biochemical mechanisms are needed to uncover the role of ALA in plants grown under water-stress conditions.

## CONCLUSIONS

Data shows that against water stress, ALA has many protective aspects to enhance plant growth and yield through improvement in pigmentation, enzymatic and non-enzymatic antioxidative mechanisms with reduction in oxidative damage and improved water relation. Among different modes of ALA treatment, the growth and yield of water-stressed mungbean plants were superior to those grown from seeds treated with ALA than foliar and combined treatment. Thus, as seed priming, ALA is more promising in combating the adverse effect of water stress, which will surely be helpful for the farmers to get better mungbean production under deficit irrigation or in rainfed conditions.

## ABBREVIATIONS

| | |
|---|---|
| **RFW** | root fresh weight |
| **RDW** | root dry weight |
| **SFW** | shoot fresh weight |
| **SDW** | shoot dry weight |
| **SL** | shoot length |
| **RL** | root length |
| **Chl. *a*:** | chlorophyll *a* |
| **Chl. *b*:** | chlorophyll *b* |
| **TSP** | total soluble protein |
| **MDA** | malondialdehyde |

| **H** | $_2O_2$ hydrogen peroxide |
|---|---|
| **SOD** | superoxide dismutase |
| **POD** | peroxide dismutase |
| **CAT** | catalase |
| **TFC** | total flavonoid content |
| **TPC** | Total phenolic content |
| **GB** | glycine betaine |
| **AsA** | ascorbic acids |

## ACKNOWLEDGEMENTS

The authors highly acknowledge the Experimental Botany Lab and Plant physiology Lab, Government College University Faisalabad, for providing the facilities and instruments for conducting the analysis.

### Funding

The authors received no funding for this work.

### Competing Interests

The authors declare that they have no competing interests.

### Author Contributions

- Naima Hafeez Mian performed the experiments, prepared figures and/or tables, and approved the final draft.
- Muhammad Azeem conceived and designed the experiments, authored or reviewed drafts of the article, supervision of this Phd project, and approved the final draft.
- Qasim Ali analyzed the data, prepared figures and/or tables, co Supervision of this phd Project, and approved the final draft.
- Saqib Mahmood conceived and designed the experiments, prepared figures and/or tables, and approved the final draft.
- Muhammad Sohail Akram conceived and designed the experiments, authored or reviewed drafts of the article, and approved the final draft.

### Data Availability

The raw data is available in the Supplemental File.

### Supplemental Information

Supplemental information for this article can be found online at http://dx.doi.org/10.7717/peerj.17191#supplemental-information.

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
