# Peer review of "Alpha lipoic acid mitigates adverse impacts of drought stress on growth and yield of mungbean: photosynthetic pigments, and antioxidative defense mechanism"

_PeerJ, doi:10.7717/peerj.17191_

## Round 0.1 · original submission · Major Revisions

Mitigative treatments to reduce the adverse impacts of the drought will be one of the most important issues that scientists may spend so much time on in the future. Therefore, your article has valuable data for the next studies. However, your article needs a comprehensive revision in many aspects such as linguistics, the meaning of some statements, and conveying your thoughts to the reader. Please carefully read the reviewers' suggestions about your article. If you do not accept one or more of their suggestions, give your reasons.

**Language Note:** The review process has identified that the English language must be improved. PeerJ can provide language editing services - please contact us at copyediting@peerj.com for pricing (be sure to provide your manuscript number and title). Alternatively, you should make your own arrangements to improve the language quality and provide details in your response letter. – PeerJ Staff

Reviewer 1 ·

Basic reporting

The manuscript titled "Ameliorating Impacts of Exogenously-Applied Alpha Lipoic Acid on Growth and Yield of Differentially Drought-Tolerant Mungbean Genotypes: Photosynthetic Pigments, Lipid Peroxidation, Antioxidative Defense Mechanism" reflects intensive work and has been well articulated. However, I have some comments that I think should be addressed to improve its status for acceptance and publication.
The article requires major revisions:

1. The abstract lacks numerical values in the key results. Providing numerical values along with the results would enhance its clarity and comprehensiveness. Implications are very general and broad. implications should be discussed based on ALA application.
2. In the introduction, Line 93, expand upon the problem in greater detail. Provide references and elucidate what sets your study apart and makes it distinct. After the objective in the last line of the introduction, authors should develop a further explanation of how they tackled that problem in the manuscript and a broader explanation.
3. It's essential to address English grammar and spelling errors throughout the article, as there are numerous incomprehensible phrases.
4. Ensure consistent and proper use of punctuation, including points, commas, spaces, and capital letters throughout the manuscript.
5. Regarding the discussion in Lines 504 and 505, update references to incorporate the latest literature on carbon metabolism, biomass, and phenolics under drought conditions. Incorporating recent literature will enrich the discussion. overall, the discussion is long and can be comprehensive.
6. The conclusion should be lucid and emphasize the novelty of your findings.
7. resolution of figured can be improved.

Experimental design

no comment

Validity of the findings

The conclusion should be lucid and emphasize the novelty of your findings.

Additional comments

no comment

·

Basic reporting

The English language needs improvement.
Literature cited is appropriate
Professional article structure is good, tables , figures are ok, data is also shared.

Experimental design

It ok.

Validity of the findings

It is quite meaningful as mungbean is summer crop which needs attention of researcher. Till date it is cultivated by farmers and such problems have not been addressed.

Additional comments

Comments given on text of body.

·

Basic reporting

The manuscript in general must be standardized, authors must review the journal's guidelines before trying to publish something.

Experimental design

Does the statistic meet the authors’ expectations?
I believe the experiment has its value, but I think it is necessary to mention where the doses came from.

Validity of the findings

No comment.

Additional comments

Nothing to declare.

---

## Round 0.2 · Minor Revisions

I want to thank you for accepting the suggestions of the reviewers. However, some issues should be completed.

Please write technical terms instead of abbreviations in the abstract.

The conclusion sentence in the abstract should be revised.

Please add which package was used in Rstudio.

Please check the lines following: 161, 162, 179, 202, 222, 237, 245, 392, and 434.

Please check the DOI numbers of the references.

Please check your title, especially (:).

There are a few grammatical and spelling errors in the article, despite your thorough review. I recommend seeking assistance from either a colleague or an editing service to refine and polish your article.

See the annotated file provided by Reviewer 3.

Reviewer 1 ·

Basic reporting

No comments

Experimental design

No comments

Validity of the findings

No comments

Additional comments

Authors answered all the queries and significantly improved the manuscript.

·

Basic reporting

Nothing to declare.

Experimental design

Nothing to declare.

Validity of the findings

Nothing to declare.

Additional comments

Nothing to declare.

---

## Round 0.3 · Minor Revisions

I would like to thank you for accepting of referees' suggestions and improving your article based on their suggestions. I think your article is almost ready to publish.

The paper needs English proofreading. For example in conclusions:
>> "Conclusively, it was found that 0.1 and 0.15 mM levels of ALA as seed
>> priming were found better to"

**Language Note:** The Academic Editor has identified that the English language must be improved. PeerJ can provide language editing services - please contact us at copyediting@peerj.com for pricing (be sure to provide your manuscript number and title). Alternatively, you should make your own arrangements to improve the language quality and provide details in your response letter. – PeerJ Staff

---

## Round 0.4 · accepted · Accept

I would like to express my appreciation for incorporating the referees' suggestions and enhancing your article accordingly. I believe your manuscript is now ready for publication. We look forward to your next article.